# Beyond Canine Adipose Tissue-Derived Mesenchymal Stem/Stromal Cells Transplantation: An Update on Their Secretome Characterization and Applications

**DOI:** 10.3390/ani13223571

**Published:** 2023-11-19

**Authors:** Barbara Merlo, Eleonora Iacono

**Affiliations:** 1Department of Veterinary Medical Sciences, University of Bologna, 40064 Bologna, Italy; eleonora.iacono2@unibo.it; 2Interdepartmental Centre for Industrial Research in Health Sciences and Technologies, University of Bologna, 40126 Bologna, Italy

**Keywords:** canine, mesenchymal stromal cells, adipose tissue, secretome, extracellular vesicles, conditioned medium

## Abstract

**Simple Summary:**

A dog is not only a patient but also a promising biomedical model suitable for the evaluation of novel therapies. Mesenchymal stem/stromal cells (MSCs) are valuable tools for the regeneration of damaged tissues in clinical applications. Apart from their plasticity and differentiation ability, it is now well known that paracrine mechanisms play a primary role in tissue regeneration. Indeed, MSCs release bioactive molecules, generally named secretome, which exert therapeutic functions. The secretome consists of many different soluble (growth factors, chemokines and cytokines) and non-soluble factors (extracellular vesicles). This review provides an update on the state-of-the-art characterization and applications of the conditioned medium or extracellular vesicles obtained from canine adipose tissue-derived MSCs.

**Abstract:**

A dog is a valuable animal model and concomitantly a pet for which advanced therapies are increasingly in demand. The characteristics of mesenchymal stem/stromal cells (MSCs) have made cell therapy more clinically attractive. During the last decade, research on the MSC therapeutic effectiveness has demonstrated that tissue regeneration is primarily mediated by paracrine factors, which are included under the name of secretome. Secretome is a mixture of soluble factors and a variety of extracellular vesicles. The use of secretome for therapeutic purposes could have some advantages compared to cell-based therapies, such as lower immunogenicity and easy manufacturing, manipulation, and storage. The conditioned medium and extracellular vesicles derived from MSCs have the potential to be employed as new treatments in veterinary medicine. This review provides an update on the state-of-the-art characterization and applications of canine adipose tissue-derived MSC secretome.

## 1. Introduction

Veterinary species have some advantages over rodents as translational models for human diseases. First, domestic animal results from in vitro experiments can be tested in vivo in the same species on spontaneously occurring diseases. Therefore, dogs are considered excellent models for human diseases, as they are companion animals whose emotional, physical, social, and behavioral needs are met by daily interactions with humans. As a result of sharing the same environments with humans, dogs develop pathological conditions similar to those in humans [1]. Furthermore, the veterinary profession has developed knowledge and expertise in the research and care of dogs, for which owners increasingly require innovative therapies, such as those promoted in the field of regenerative medicine. In this context, the information obtained from the studies on dogs may be relevant not only for the applications in this species but also for translational purposes [2].

Over the past decade, stem cell research has been highly active, and cell-based therapies have advanced as tools for regenerative medicine due to the increasing number of studies and the breakthroughs achieved in this field. Mesenchymal stem/stromal cells (MSCs) can be found in various body tissues, are able to adhere to plastic, express specific surface antigens, and possess a multipotent differentiation potential [3]. Furthermore, their low immunogenicity, anti-inflammatory potential, and ability to release several mediators and bioactive molecules that are involved in regenerative functions make them promising candidates for the treatment of various diseases [4].

MSCs can differentiate into cells originating from the mesoderm, such as adipocytes, osteocytes, and chondrocytes, as well as into cells from endodermal or ectodermal lineages, such as hepatocytes or neurocytes [5]. Nonetheless, the trans-differentiation of MSCs from the original mesoderm into cells of ectodermal origin remains controversial [6]. Even so, the multilineage differentiating ability of MSCs expands the possibilities of regenerative medicine [7], in which MSC applications appear as potentially valuable remedies for bone and cartilage defects [8,9].

On the other hand, although their therapeutic effect was attributed initially to engrafting and differentiation, it was later demonstrated that transplanted MSCs have low survival in host tissues, and their retention is only transient [10]. These findings suggested that the therapeutic effects of MSCs are exerted by the secretion of bioactive factors that induce a beneficial microenvironment to promote the restoration and regeneration of damaged tissues [11]. The bioactive molecules or growth and trophic factors produced by MSCs in the target tissue affect immunoregulation, angiogenesis, and neuroprotection through a paracrine mechanism [12,13].

The complexity of MSC secretome is not fully understood yet, and MSC secretome-based therapy in human and veterinary medicine still has to be fully optimized and developed [14,15]. The first obstacle when considering using MSC secretome as a therapeutic option is its intrinsic variability, since the donor, cell tissue origin, passage of the culture, and the culture conditions may influence its composition [14,16]. To develop customized therapies for specific tissue injuries, a better understanding of the bioactive factors produced by MSCs derived from different tissue sources is needed [17].

Nonetheless, cell-free therapies have several advantages compared to cell-based therapies when considering the risk factors associated with stem cells use. There are intrinsic factors related to the cell origin, tumorigenic potential, proliferation and differentiation capacity, as well as extrinsic factors such as cell handling, storage, and transportation [18]. Secretome-based therapies (CM or EVs) have a lower risk of thrombosis after intravenous administration [19], no risk of unlimited cell growth or tumor formation, and no undesired consequences after the genetic modification of the cells [20], in addition to easier manufacturing, handling, delivery, storage, and lower costs [17]. In fact, similar to pharmaceutical agents, secretome may be evaluated for its safety, dosage, and potency and could be mass produced under controlled laboratory conditions, tailoring the composition to obtain the desired therapeutic actions, and stored for ready-to-use applications in acute conditions without the need for a donor and the time required for cell expansion [17].

Even though canine MSCs have been isolated from many tissues, the prominent role of adipose tissue (AT) in cellular therapy has become evident [21]. In a comparative study [22], canine AT-MSCs (cAT-MSCs) showed better results in terms of the yield of isolation, phenotype homogeneity, multilineage potential, and proliferation capacity compared to the MSCs derived from bone marrow and the amniotic membrane. Since the isolation and first characterization of cAT-MSCs [8], several studies have analyzed the different possibilities for using these cells, alone or in combination with other treatments, as therapy for various canine pathological conditions [21] (Figure 1). In addition to the unique properties of MSCs, their secretome has recently drawn attention. This review provides an update on the state-of-the-art characterization and applications of cAT-MSC secretome.

## 2. Secretome Characterization

The term secretome was firstly coined as a reference to all secreted proteins and the secretory machinery of bacteria [23]. In broad terms, secretome is defined as the total amount of molecules/factors released by the cell into the extracellular space. Each cell and tissue produce a specific secretome, with changes occurring in relation to physiological or pathological stimuli. Among the factors produced, free nucleic acids, soluble proteins, lipids, and extracellular vesicles (EVs) are included.

### 2.1. Extracellular Vesicles

EVs are made of lipid bilayers containing various biomolecules, such as lipids, proteins, and nucleic acids, which show different compositions in relation to their biogenesis [24]. The EVs’ membrane lipid fraction contains cholesterol, sphingomyelin, and phospholipids in varying proportions [25]. Additionally, several proteins are present for adhesion, cell communication, and trafficking. Among the proteins enclosed by EVs, enzymes, cytoskeleton proteins, and signaling proteins have been reported [25]. Moreover, EVs encapsulate a wide variety of RNA molecules and fragments of DNA.

EVs can circulate in the blood and reach cells that are distant from their origin. Different mechanisms have been reported for their interactions with target cells. They can fuse with the plasma membrane and directly release their content into the cell cytosol, be internalized after binding the receptors on the cell membrane, or enter the cell through endocytosis [26].

EVs have been studied as important immunomodulatory messengers in the context of tissue repair and regeneration [27]. They participate in a physiological strategy of cell communication to regulate immune cell functions and influence the processes involved in inflammation resolution during repair and, subsequently, in tissue regeneration [27].

EVs are broadly classified into different subpopulations. Following the indications of the International Society for Extracellular Vesicles (ISEV) according to the Minimal Information for Studies of Extracellular Vesicles 2018 (MISEV2018) [28], since there is a lack of consensus on the specific markers of the EVs subtypes, they should be named and classified by considering their physical characteristics, such as their size or density, biochemical composition, or descriptions of the conditions or cell of origin (Figure 2).

Despite the therapeutic and biological application possibilities, few studies have characterized EVs derived from cAT-MSCs. The following table summarizes the existing research (Table 1).

The MISEV 2018 [28] update outlined the steps for EVs characterization, providing protein categories for consideration and making recommendations. The proteins labeled for evaluation in any EV preparation included (i) those associated with the presence of a lipid bilayer in the material analyzed, (ii) cytosolic/periplasmic proteins with a lipid or membrane protein-binding ability, and (iii) purity controls. The typically established markers were proteins found in mammalian cell-derived EVs, so they were not species-specific.

In the canine context, the first study characterizing cAT-MSC EVs dates back 2019 [29]. Exosomes were examined using transmission electron microscopy (TEM) to determine their size and shape and Western blotting (WB) for the expression of the specific exosomal surface protein markers involved in exosome biogenesis mediated by the endosomal sorting complexes required for transport. A total of 47 proteins were identified in the cAT-MSC exosomes (28 characterized and 19 uncharacterized) with various biological functions, including cell communication, differentiation, organization and biogenesis, cellular homeostasis, transport, metabolic processes, regulation of biological processes, and response to stimuli [29]. Given the growing interest in the role of exosomes as therapeutics for various diseases, their immunomodulatory potential was also investigated. The ability to inhibit the proliferation of peripheral blood mononuclear cells (PBMCs) was assessed in vitro for both MSCs (1 × 10^4^), their conditioned mediums (CMs 150 μL), and their exosomes (20 μg/mL) [29]. The cAT-MSCs and their CM suppressed T-cell activation, while the exosomes did not exhibit the same immunomodulatory capacity at the concentration used [29].

The EVs derived from primed cAT-MSCs (with tumor necrosis factor alpha: TNF-α, and interferon gamma: INF-γ) were also characterized [30]. While they were morphologically similar to naive EVs, their size ranged between 30 and 120 nm and they were predominantly round shaped [30]. The WB analysis revealed that all the EVs expressed exosomal markers (tetraspanin family of proteins), such as CD63 and CD9 (CD: cluster of differentiation), and low concentrations of the cytosolic marker β-actin (a negative marker for EVs) [30]. In another study on cAT-MSCs primed with TNF-α and INF-γ, the size and concentration of the naive EVs were 166 ± 7.7 nm and 34.4 ± 3.1 × 10^9^ particles/mL, and those of the primed EVs were 145 ± 1.5 nm and 42.8 ± 1.9× 10^9^ particles/mL, as determined using NTA (nanoparticle tracking analysis) technology [33]. Both the naive and primed EVs expressed specific exosomal markers, namely CD9, CD63, and TSG101 (tumor susceptibility gene 101, involved in exosome biogenesis and secretion), and 70 exosome-associated microRNAs (miRNAs) [33]. Furthermore, adding the primed EVs to lipopolysaccharide (LPS)-stimulated DH82 cells (canine macrophages) resulted in a decrease in the expression of TNF-α and pro-inflammatory interleukins (IL-1β, and IL-6), and an increase in the IL-10 (with both pro- and anti-inflammatory actions) levels [30]. The iNOS (inducible nitric oxide synthase, which promotes the differentiation of the M1 phenotype) levels were reduced, while the CD206 and arginase (both promoting the M2 phenotype) levels increased [30]. Overall, the results suggested that the EVs derived from TNF-α and INF-γ primed cAT-MSCs induced the M2 macrophage phenotype more effectively than the naive EVs [30]. In addition, the similarly primed EVs exerted immunosuppressive effects on the stimulated CD4^+^ T cells in vitro [33].

To elucidate the role of TSG-6 (TNF-α-stimulated gene/protein 6, which is secreted from stem cells and is a key factor in regulating inflammatory responses) in EVs for mitigating inflammation, cAT-MSCs were transfected with small interfering RNA (siRNA) [31]. TEM revealed that the EVs were round shaped and 50–100 nm in diameter, which was confirmed by a particle size analyzer [31]. The positive markers of the EVs (CD63 and CD9) were identified using WB, whereas the negative markers of the EVs (laminin A, a nuclear marker, and β-actin, a cytosolic marker) were present at a lower abundance [31].

The EVs separated from the cAT-MSC culture media by Exo-quick™ were round shaped and ranged from 50 nm to 100 nm in diameter using TEM, which was confirmed to be around 100 nm using a particle size analyzer [32]. The surface markers were analyzed using WB, which confirmed the presence of CD81 and CD9 and the absence of β-actin [32]. Additionally, in this study, an attempt was made to increase the secretion of anti-inflammatory factors by priming the cAT-MSCs with hypoxia or a hypoxia-mimetic agent (deferoxamine: DFO) to produce EVs enriched with the COX-2 (cyclooxygenase-2) protein. Their anti-inflammatory effect was verified on the LPS-stimulated DH82 cells, and the COX-2-enriched EVs were able to promote the phosphorylation of STAT3 (signal transducer and activator of transcription 3), reprogramming macrophages into the M2 phase [32].

In another study, the EVs were isolated using ultrafiltration and the particle size distribution and characterization were evaluated using NTA [34]. The vesicle diameter ranged between 190–230 nm, encompassing the exosomes (40–120 nm) and microvesicles (250–1000 nm). Furthermore, a heterogeneous population was shown by analyzing the curve, with d10 between 106–122 nm, d50 between 157–180 nm, and d90 between 316–400 nm [34].

In another study using tangential flow filtration for EV isolation, the size of the exosomes was around 182 nm [35]. Alix, flotillin, and CD9 (exosomal-specific markers) were detected using WB. Several surface CD molecules, including CD105, CD49e, CD9, CD41b, CD81, CD44, and CD29, were markedly expressed in the EVs. However, they did not contain human leukocyte antigens (HLA-DRDPDQ and HLA-ABC) and co-stimulatory molecules (CD80 and CD86), suggesting their low immunogenicity [35].

In a more recent study using the same isolation method, the EV size was around 104 nm, they were positive for CD81, and had lower levels of calnexin compared to the cell lysate, which confirmed their identity [36]. Additionally, 798 miRNAs in AT-MSC EVs were identified using next-generation sequencing (NGS). Some of the top seven miRNAs (let-7a, let-7b, miR-21, let-7f, miR-125b, miR-24, and miR-29a), accounting for 66% of the total miRNAs, are known to be involved in the regulation of inflammation [36]. Particularly, let-7a, let-7b, and miR-21 have anti-inflammatory actions through miRNA transfer into the recipient cells [37,38,39]. According to the Gene Ontology (GO) analysis of the top 20 target miRNAs, they were related to the intrinsic apoptotic signaling pathway in response to DNA damage (biological process), protein serine/threonine kinase activity (molecular function), and the JAK-STAT signaling pathway [36]. Serine/threonine kinase play key roles in cellular growth, differentiation, and secondary metabolism by sensing changes in the external environment and altering the gene expression to trigger the proper metabolic processes [40]. The JAK/STAT pathway is involved in the regulation of embryo development and in the control of hematopoiesis, inflammatory response, and stem cell maintenance [41].

The characterization of the miRNA cargo of cAT-MSC EVs is a first step for understanding their therapeutic bioactivity and potential. Indeed, the contents of EVs is dependent on the cell source, as demonstrated by profiling the miRNA and protein cargos of the EVs derived from three different human stem cell-based products (bone marrow-derived MSCs, heart-derived MSCs, and umbilical cord-derived MSCs) [42]. Several common features were present, but functional differences were found [42]. On the other hand, the application of biologicals faces a crucial limiting factor related to the delivery method of bioactive molecules to the target site. New powerful therapeutics, such as RNA, are unable to cross the cell membrane. EVs can act as mediators for intracellular signaling by transporting bioactive molecules and, depending on the type, inducing therapeutic responses. Consequently, the field of EVs research is rapidly growing since their therapeutic potential is interesting for both drug delivery and regenerative medicine. The natural and engineered cellular uptake and trafficking of EVs have been studied, but many aspects have to be elucidated before they can be successfully transferred into therapeutics [43].

### 2.2. Soluble Factors

Various soluble factors, primarily cytokines, chemokines, growth factors, and extracellular matrix components, are part of the complex mixture that, in conjunction with EVs, constitutes the secretome produced by MSCs. Similar to EVs, these factors play a crucial role in understanding the therapeutic potential of MSCs. However, the paracrine profile of MSC secretome has been limitedly studied in the veterinary field.

In dogs, the secretory profile of cAT-MSCs was analyzed for the presence of chemokines (monocyte chemoattractant protein-1: MCP-1), cytokines (interleukins: IL-2, IL-6, IL-8, IL-10, IL-12p40, TNF-α, and IFN-γ), immune mediators (prostaglandin E2: PGE2, nitric oxide: NO, indoleamine 2,3-dioxygenase: IDO), and growth factors (beta-nerve grown factor: NGF-β, stem cell factor: SCF, transforming growth factor beta: TGF-β, vascular endothelial growth factor A: VEGF-A) [29]. The most secreted component was MCP-1, a chemokine that attracts or enhances the expression of other inflammatory cells and factors and plays a key role in the inflammatory process [44]. Five pro-inflammatory cytokines (IL-2, IL-6, IL-8, INF-γ, and IL-12p40) and IL-10, an immunomodulatory cytokine with both pro- and anti-inflammatory actions, were all detected [29]. Regarding the growth factors, the CM contained [29] various levels of VEGF-A, which mediated angiogenesis; NGF-β, a target-derived protein necessary for the survival and development of sympathetic and sensory neurons [45]; SCF, a hematopoietic cytokine that plays a role in the preservation of the viability of hematopoietic and progenitor cells and in the facilitation of their proliferation and differentiation [46]; and TGF-β, a tumor suppressor. Other mediators in the immunomodulatory capacity of MSCs were differently detected. The IDO activity was not shown, PGE2 was strongly expressed, and NO was almost absent [29]. These results were in line with what was observed by studying the mechanisms of immune suppression utilized by canine MSCs.

Since dogs develop inflammatory and autoimmune diseases that can be considered spontaneous models for human diseases, knowing which immunomodulation mechanisms are operative when using MSC therapy is useful for comparative studies. IDO is a catabolic enzyme protein involved in tryptophan metabolism that inhibits the metabolism in various biological systems, including stem cells [47]. As reviewed in [47], human MSCs do not possess an innate ability to express IDO, but when they are stimulated by a combination of pro-inflammatory cytokines, they gain this ability. Furthermore, distinct species have considerable variations in MSC-mediated immunosuppression, as human and monkey MSCs utilize predominantly IDO as an immunosuppressive molecule, while mouse MSCs use NO [47]. In dog AT-MSCs co-cultured with leukocytes, the immunomodulatory factors such as TGF-β, the hepatocyte growth factor (HGF), PGE2, and IDO were increased. However, leukocyte proliferation was successfully restored only by the IDO and PGE2 inhibitors, even though the IDO and PGE2 production followed different kinetics [48]. Then, to better define the functional immunomodulatory properties of canine MSCs, different pathways of the MSC suppression of T cells already observed in other species were investigated in dogs as well [49]. It was observed that cAT-MSCs did not use the NO pathway for T cell suppression, and, in contrast with the previous results, the IDO pathway did not play a key role in immunosuppression [49]. In fact, the cAT-MSCs utilized the TGF-β signaling pathway and adenosine signaling to suppress T cell activation [49].

The secreted proteins in the CM derived from cAT-MSCs were identified using the iTRAQ proteomic analysis [50]. This analysis revealed 1910 proteins that were categorized as being involved in biological process (72.9%), molecular and various related functions (17.1%), and cellular components (10.0%), according to the GO analysis [50]. The levels of several secreted cytokines and growth factors (IL-1β, IL-2, IL-6, IL-8, IL-10, IL-12p40, IL-17A, the granulocyte–macrophage colony-stimulating factor: GM-CSF, MCP-1, the receptor for advanced glycation end products: RAGE, SCF, TNF-α, VEGF-A, erythropoietin: EPO, the keratinocyte growth factor: KGF, the HGF, the hepatocyte growth factor receptor: HGF-R, IFN-γ, macrophage inflammatory protein 1 beta: MIP-1β, and the tumor necrosis factor receptor I: TNF-RI) were also determined. Only the GM-CSF was not detected [50]. Consequently, it was demonstrated that the secretome of canine AT-MSCs served as a rich source of soluble factors (Figure 3), which were involved in the establishment of an appropriate microenvironment within the injured tissue, promoting cell survival, regeneration and differentiation, immunomodulating the inflammatory response, and stimulating angiogenesis.

Furthermore, considering the potential of MSC-secretome as a substitute for MSCs, a good manufacturing practice (GMP)-compliant human secretome production process was established to produce a stable freeze-dried secretome referred to as “Lyosecretome“. The same process was subsequently applied to obtain Lyosecretome from cAT-MSCs [34]. Injectable canine Lyosecretome was prepared under ISO9001 clinical grade conditions. The ultrafiltrate was specially prepared to contain both EVs and proteins with a molecular weight exceeding 5 kDa. This aspect was functional to the production of a pharmaceutical product, which was characterized for the total protein and lipid content, physical–chemical properties, and particle size [34]. However, qualitative analyses of the protein, RNA, and lipid contents were lacking. Nevertheless, it was highlighted that there was variability in the protein and lipid content depending on the cell line used, and the in vitro tests were used to confirm the potency and efficacy of Lyosecretome [34]. It was found to stimulate the metabolic activity of the target cells (canine tenocytes, chondrocytes, and AT-MSCs) in a dose-dependent manner, and the same dose-related trend was observed for its anti-elastase activity [34].

Overall, the research on the soluble factors characterizing canine secretome derived from cAT-MSCs remains an evolving field. The wide individual variability, which is a characteristic trait of both the cells and their secretions, certainly presents a complex challenge. Conversely, in vitro and in vivo applications of the CM or EVs have yielded promising results, advancing the understanding of the factors and mechanisms that underlie their effects.

## 3. Secretome Applications

Secretome is rich in molecules that are responsible for the therapeutic effects of MSCs. It is easy to produce, scale, and store, and its administration in experimental disease models has been demonstrated to be as effective as treatment with MSCs. Furthermore, its composition could potentially be customized in the future by exposing MSCs to various stimuli under controlled laboratory conditions to enhance the secretion of target molecules, which are therapeutically valuable for a specific disease. However, research is needed to select the most appropriate cell source based on the composition of its secretion in relation to the treated pathology. Human MSC secretome has been largely investigated, but despite the interesting results, no MSC secretome-based therapy has been approved for human medical use [2]. A dog is certainly an interesting preclinical model and a patient itself. Their natural disease incidence, similarity in tissue biology, and shared living environments make dogs attractive surrogates for human studies. Overall, the information obtained from studies on canine MSCs can be valuable not only for veterinary medicine but also for relevant translational applications for human conditions.

The in vitro and in vivo applications of the CMs or EVs derived from cAT-MSCs for treating various pathologies are chronologically summarized in Table 2. Due to the presence of multiple areas of application, the details of the studies will be presented below by grouping the pathologies into broader fields. The review will focus on the secretome (CMs or EVs) applications and results, even when MSCs have been concurrently used in the experimental designs.

### 3.1. Inflammatory and Immuno-Mediated Diseases

Inflammatory bowel disease (IBD) is a chronic gastrointestinal tract disease characterized by inflammation resulting from incorrect responding mechanisms of the immune system. IBD affects both dogs and humans, negatively influencing their quality of life and longevity. Typical pharmacological treatments involve the use of antibiotics and immune suppressors. However, recurrence is frequent, thus leading to a search for new therapeutic approaches [61,62].

One strategy investigated in dogs for the management of IBD is the use of cAT-MSC EVs [30,31]. To enhance the anti-inflammatory and immunosuppressive potential of the EVs, pro-inflammatory cytokines such as TNF-α and IFN-γ were used to prime the cAT-MSCs [30]. The in vitro stimulation increased the presence of immunosuppressive proteins, such as the HGF, TSG-6, PGE2, and TGF-β, in the derived EVs and the resulting anti-inflammatory effect [30]. Indeed, the primed EVs were able to induce M2 macrophage polarization (anti-inflammatory phenotype) in vitro (100 μg EVs/well) and in vivo (mice model, 100 μg EVs/mouse) and to enhance regulatory T cells and regulate the M1/M2 balance in the inflamed colon, thereby suppressing the activated immunity [30]. The same research group hypothesized that TSG-6 in the EVs could play a key role in alleviating colitis symptoms. Thus, they used dextrane sulfate sodium (DSS)-induced colitis mice to evaluate the underlying mechanisms using TSG-6-containing/depleted EVs [31]. They observed that the TSG-6 depleted EVs exhibited a reduced anti-inflammatory action compared to the TSG-6-containing EVs, emphasizing the role of TSG-6 in macrophage polarization from M1 to M2 and in increasing regulatory T cells in the colon [31].

These studies provided a basis for further research into the potential application of cAT-MSC-derived EVs in the treatment of IBD and immune-mediated diseases in dogs. Moreover, given the spontaneous occurrence of canine IBD, it may be considered an appropriate model for future translational research.

### 3.2. Infections

Bacterial resistance to antimicrobials is a growing issue for both human and animal health [63,64]. Drug-resistant infections represent a challenge that requires the development of new treatment strategies. Along with the increasing interest in MSC-based therapies, concerns have arisen regarding the potential increased risk of infection due to an unwanted suppression of antimicrobial immunity. On the contrary, it has been demonstrated that MSCs exhibit antimicrobial properties via both direct (locally acting antimicrobial peptides) and indirect (systemic activation of host innate immune effector cells) means, which are partially mediated by the release of antimicrobial peptides and proteins (AMPs). Notably, MSCs can enhance the bacterial clearance in preclinical models of sepsis, acute respiratory distress syndrome, and cystic fibrosis-related infections (as reviewed by [65]).

The antimicrobial properties of AT-MSCs have also been investigated in dogs. Their potential direct and indirect mechanisms explaining the bactericidal activity and their interactions with common classes of antibiotics were explored after activation by a TLR3 (Toll-like receptor 3) agonist [58]. To evaluate the efficacy of the activated cAT-MSCs as a new tool for the clinical management of drug-resistant infections, a trial on dogs with naturally occurring chronic drug-resistant infections was performed, in addition to in vitro tests performed on the CM produced by the activated cells [58]. The latter showed that the CM from the activated cells did not exert an effective direct bactericidal activity. However, when combined with the sub-therapeutic antibiotic concentrations, there was an additive or synergistic bactericidal activity with all the major classes of antibiotics tested, and with both antibiotic-resistant *Staphylococcus* and *E. coli* [58]. In detail, bactericidal synergy was demonstrated for aminoglycosides (gentamicin), fluoroquinolones (enrofloxacin), cephalosporins (cefazolin), vancomycin, and the bacteriostatic chloramphenicol [58]. Moreover, the indirect effects of activated MSC secretome on canine macrophage bactericidal and functional properties were investigated. The CM from the activated cAT-MSCs was found to enhance the macrophage bactericidal activity after 48 h of co-incubation [58]. Analyzing the macrophage polarization after exposure to the CM, a slightly greater M1 phenotype was observed compared to direct exposure to the activated cells. However, no secretion of TNFα, which is generally associated with an M1 or inflammatory macrophage, was detected [58]. The potential for stimulating MSCs to synthesize and release specific molecules, thereby tailoring the therapeutic potential of their secretome, opens a broad area of investigation.

The synthesis of new antimicrobials and AMPs represents a valuable approach for facing drug resistance issues. However, to overcome infections, other strategies such as improving natural defense mechanisms, AMP pharmacokinetics, and the antimicrobial actions are required. From this perspective, MSCs and their secretome can be considered promising tools for defeating antimicrobial resistance [65].

### 3.3. Skin and Ear Diseases

Canine atopic dermatitis (AD) is a progressive, pruritic, chronically relapsing, genetically predisposed, and inflammatory skin disorder [66]. Similar to other pathologies, canine AD shares resemblances with human AD, including the clinical signs, immunopathological characteristics, therapeutic approaches, and treatment responses, making dogs a suitable spontaneous animal model [66]. Although the pathogenesis of AD is still not completely understood, it entails epidermal barrier dysfunction, immune dysregulation, and dysbiosis of the cutaneous microbiome resulting from complex interactions between immune functions and environmental (allergens) and genetic factors [66,67]. Since pruritus is the main clinical sign and can be complicated by various flare factors, the management of AD involves the use of a combination of long-term therapies. Depending on the severity of the pathology, topical and systemic antipruritic and anti-inflammatory drugs are usually required in addition to tailored therapies [66]. Progress in the study of new therapies is crucial for developing effective, long-term treatments with an acceptable safety profile, especially in the moderate to severe forms [67].

EVs derived from cAT-MSCs have been used to treat AD [35,36]. In the first study, to validate the therapeutic potential of cAT-MSCs and their EVs, AD was induced in a mouse model using the topical 1-chloro-2,4-dinitrobenzene treatment. Interestingly, both the MSC (2 × 10^6^ cells/head) and EV (2 × 10^10^ particles/head) treatments effectively alleviated induced dermatitis; suppressed epidermis hyperplasia, parakeratosis, and dermal oedema; and decreased the levels of the serum IgE [35]. Moreover, they ameliorated inflammation by suppressing the immune response to AD through the reduction in mast cell infiltration and the production of Th2-related chemokines and cytokines [35]. Concerning the skin barrier function, hydration and the trans-epidermal water loss (TEWL) were measured, and the expression levels of the epidermal differentiation proteins (keratin 1, filaggrin, loricrin, and involucrin) were analyzed. It was found that both the AT-MSCs and derived EVs promoted skin barrier repair by reducing the TEWL, improving hydration, and increasing the expression of the proteins involved in epidermal differentiation [35]. Both treatments were also effective in suppressing IL-31/TRPA1 (transient receptor potential ankyrin 1)-mediated pruritus and phosphorylated STATS (STAT1 and STAT3) [35].

In the second study, a Biostir-induced AD mouse model was used to test the cAT-MSC and EV SC treatment, and a toxicity study was also performed [36]. The EVs improved skin lesions and AD symptoms and reduced the serum levels of IgE, inflammatory cytokines and chemokines, and ear thickness in a dose-dependent manner [36]. No systemic toxicity was observed when administering a single (low dose 7.45 × 10^8^, mid-dose 2.98 × 10^9^, and high dose 1.19 × 10^10^ particles/20 g) or repeated-dose (three times a week), confirming the safety of this cell-free therapeutic option [36]. The highest dose used in the study was considered to be the NOAEL (no observed adverse effect level) of the cAT-MSC EVs [36]. In the same study, EV-miRNAs were investigated, confirming that miRNA transfer through the EVs plays a role in the intracellular communication network that orchestrates immune responses [36].

Another skin disorder affecting the patient’s quality of life is related to wound healing. The heterologous therapeutic potential of cAT-MSC secretome was investigated in turtles (*Trachemys scripta*), where the healing process was lengthier compared to mammals [59]. In this case, the treatment did not improve the healing process with the dose that was used. Although the xenogeneic biological activity of MSCs has been demonstrated [68], the phylogenetic distance between mammals and reptiles might have been responsible for the lack of activity of the secretome components, or a higher dose might have been required [59]. Little is known about the utilization of xenogeneic MSC secretome between phylogenetically distant species.

Like skin, the epithelial cells within the tympanic membrane (TM) and the external ear canal can regenerate. These regenerating cells migrate outward from the TM toward the opening of the ear canal [69], preventing the accumulation of cerumen, desquamated substances, and debris in the external acoustic meatus. Furthermore, epithelial migration (EM) is involved in wound repair since squamous migrating cells can reach and cover the perforation, closing the defect before fibrous tissue advancement [70]. Therefore, an efficient EM helps maintain a healthy ear canal environment, repair TM perforations, and protect from external infectious agents. Although autologous grafts are successful in repairing TM perforation, alternative therapies are required for treating chronic unresponsive perforations and addressing the shortage of tissues for transplantation, making regenerative medicine strategies highly attractive [71,72]. With the purpose of increasing the rate of EM, thus maintaining a healthy and normal ear function, the CM from cAT-MSCs has been employed in normal beagle dogs [60]. The weekly application of the CM effectively increased the EM rate over the TM without side effects [60]. The levels of the bFGF (basic fibroblast growth factor), but not Il-6 and VEGF-A, were higher in the CM derived from the cAT-MSCs compared to that from the canine progenitor epidermal keratinocytes CM [60]. It was likely that the increased migration ability was stimulated by the presence of numerous factors in the cAT-MSC CM. Further studies are needed to test the therapeutic potential of the cAT-MSC CM on perforated TMs.

### 3.4. Orthopedic Diseases

Osteoarthritis (OA) affects humans and animals worldwide. Particularly, it is the most common canine joint disorder. OA is a chronic degenerative joint disease characterized by the loss and dysfunction of the articular cartilage, thickening of the joint capsule, and osteophytosis, leading to pain and lameness [73]. OA in dogs occurs secondarily to developmental orthopedic diseases, such as hip dysplasia, or can be related to other risk factors, such as genetics, increased body weight/condition, age, or neuter status [74]. The pharmaceutical treatment options available to manage the pain arisen from OA include non-steroidal anti-inflammatory drugs, piprants, paracetamol and paracetamol/codeine, anti-nerve growth factor monoclonal antibodies, opioids, gabapentinoids, N-methyl d-aspartate receptor antagonists, cannabinoids, tricyclic antidepressants, corticosteroids, structure modifying OA drugs, and regenerative therapies [75]. The use of MSCs in clinical OA trials have shown some beneficial outcomes, positioning MSCs as a potential alternative or adjunctive treatment option when conventional drugs do not yield adequate effects [75].

Two studies examined the effects of cAT-MSC secretome on elbow [56] or knee/elbow [34] joint OA. The CM from allogeneic cAT-MSCs was intra-articularly administered to six Labrador Retriever dogs with a non-specific diagnosis of bilateral elbow OA [56]. The efficacy of the treatment was determined by analyzing the range of motion (ROM) and the synovial fluid (IL-6, IL10, IL-8, IL-2, IL-12, TNF-α, IFN-γ, matrix metalloproteinase-3: MMP-3, and tissue inhibitor metallopeptidase 1: TIMP-1) on days 0, 14, and 42. The kinematic analysis showed an improvement in the ROM between day 0 and 42, and the concentration levels of IL-6, TNF- α, and MMP-3 decreased while the TIMP-1 levels increased on day 42 compared to day 0 [56]. In this study, both elbow joints were treated with the CM without the use of a placebo control group, and the number of dogs was limited with a relatively short follow-up period. Nevertheless, despite the shortcomings, the preliminary results obtained after double intra-articular injections of allogeneic cAT-MSC CMs showed an enhancement in the functional ability of the dogs without adverse effects [56].

In the other study, Lyosecretome (freeze-dried cAT-MSC secretome) was evaluated on dogs affected by naturally occurring OA to prove its in vivo safety [34]. For the trial, five dogs presenting bilateral knee or elbow OA were enrolled. The dogs were treated with two intra-articular injections of Lyosecretome (right joint) or placebo (left joint) in 1 mL hyaluronic acid at 40-day intervals [34]. After each treatment, a transitory reluctance to trot or gallop for two days was observed in all the patients, but it was not possible to attribute this to one specific treatment since both joints were concomitantly injected [34]. On the other hand, no systemic adverse reactions were detected. The clinical data following the examination at 40 and 80 days after the first injection showed no differences in terms of lameness and pain worsening [34]. The results obtained confirmed the safety of Lyosecretome, but the number of enrolled dogs was limited, making it insufficient to draw conclusions about the efficacy of the product [34].

Overall, similar to IBD, these studies have laid the groundwork for further research into the potential application of cAT-MSC-derived secretome in the context of OA.

### 3.5. Neurologic Diseases

Spinal cord injuries (SCI) are debilitating neurologic disorders resulting from direct damage to the spinal cord itself or from injury to the surrounding tissue and bones. Depending on the location and extent of the damage, temporary or permanent changes in movement, feeling, strength, and bodily functions may develop. The injury can trigger a complex pathophysiological cascade, including inflammatory and other processes, ultimately leading to scar formation [76]. Although different therapeutic approaches have been tried, SCI research still requires progress in developing new and combined therapies to improve damage repair and alleviate functional loss. One key area needing investigation is the development of methods for enhancing the intrinsic regenerative capacity of injured axons [76]. Indeed, the transplantation of AT-MSCs in various SCI animal models has shown beneficial effects on immunomodulation, inflammation, and the promotion of re-myelination and axonal growth [77,78,79,80].

Focusing on the studies regarding the use of cAT-MSC secretome, treating the SH-SY5Y neuronal cell line with a serum-free cAT-MSC CM increased cell proliferation, neurite outgrowth, and immune positivity for the neuronal marker βIII-tubulin [52,55]. Furthermore, the treatment of the EA.hy926 endothelial cell line with the same CM increased cell proliferation, migration, the rate of closure in a scratch assay, and induced the formation of tubule-like structures in a Matrigel assay [52,55].

Another in vitro approach for studying SCI involves using ex vivo spinal cord slice cultures (SCSCs), a well characterized interface static method [81]. The SCSCs from CD1 mice were prepared, pre-cultured for 4 days (equilibration) to allow the cells to reach stability, and then cultured in either a control medium or cAT-MSC CM [57]. The number of PI-stained nuclei was 39 ± 17/mm^2^ in the control medium vs. 17 ± 11/mm^2^ in the CM, indicating a sort of neuroprotective effect in the CM [57]. Moreover, the presence of the CM enhanced GFAP (glial fibrillary acidic protein) astrocytic immunostaining and the astrocytic process length, suggesting a positive effect on glial cell survival and reactivity [57]. Surprisingly, the CM did not affect the NG2 (neuron–glial antigen 2) levels, a marker of CSPGs (chondroitin sulphate proteoglycans) in the glia [57]. It was hypothesized that there may have been a loss of CSPGs into the surrounding medium, CSPG cell-mediated degradation, or a later increase in NG2 after the SCI [57]. Another interesting observation was the decrease in βIII-tubulin immunoreactive neurites [57]. Different possibilities were considered, namely that (i) the preparation of SCSCs may have involved a lower survival of spinal neurons during the equilibration period, and the CM was unable to protect them from death; (ii) the SCSC microenvironments may not have been permissive for neuronal maturation; and (iii) the CM could have contained bioactive molecules that caused the differentiation in neural precursors towards the glial cell lineage [57]. Calcium imaging showed that the signaling frequency of single active cells increased, even when the overall cell activity was not different [57]. The SCSC model provided different results compared to the in vitro cell cultures, but these observations suggested that the cAT-MSC CM contained not-yet-identified neurotrophic factors, and their effects may depend more on systemic factors than on local ones [57].

MSC secretome contains EVs and various molecules that play a role in the neuroprotection and immunoregulation of the nervous system. Although the mechanisms underlying these properties are still poorly understood, an in-depth study of the role of secretome and its beneficial effects on neurological diseases could contribute to the development of innovative therapeutic approaches.

### 3.6. Endocrine Diseases

Diabetes mellitus (DM) is an endocrine metabolic disease caused by alterations in insulin secretion and/or action. Type 1 diabetes mellitus (T1DM) is characterized by an absolute deficiency in insulin secretion due to an autoimmune destruction of pancreatic β-cells, while type 2 diabetes mellitus is caused by insulin resistance or islets β-cells dysfunction. Most canine DM patients suffer from T1DM, which usually can be successfully managed with exogenous insulin supplementation. However, poorly managed T1DM or cases of insulin resistance require expensive treatments with a bleak prognosis. In fact, traditional treatments counteract the symptoms, but do not prevent the failure of pancreatic β-cells or the complications of DM. MSC therapy has emerged as a new therapeutic option, and MSC exosomes have shown promising results in the treatment of DM, exhibiting a potential to become a valuable therapeutic strategy in the future [78].

In dogs, the cAT-MSC CM was used on an in vitro-induced insulin resistance model to investigate its effects and to elucidate the possible mechanisms involved [53]. The 3T3-L1 pre-adipocytes were induced to differentiate the adipocytes, which were then treated with TNF-α and hypoxia for 24 h to induce insulin resistance [53]. The insulin receptor substrate 1 (IRS-1) and insulin-regulated glucose transporter (GLUT4) were used as the markers of insulin resistance, and both showed an increased expression after treatment with the CM [53]. Given that the recombinant fibroblast growth factor 1 (rFGF-1) was demonstrated to normalize blood glucose levels and improve insulin resistance in diabetic mice [82], its role in the therapeutic effect of the CM was evaluated. When an anti-FGF-1-neutralizing antibody was added to the CM-treated insulin resistance model, the IRS-1 and GLUT4 expression levels decreased and the glucose uptake was reduced [53]. These results suggested that the cAT-MSC CM can ameliorate in vitro-induced insulin resistance and that the FGF-1 can mediate the beneficial effects of the cAT-MSC CM [53].

Various cytokines and growth factors other than the FGF-1 may be involved in the positive effect of the cAT-MSC CM on insulin resistance. Further studies are necessary to identify the other effective factors and to test the effectiveness in vivo.

### 3.7. Tumors

MSCs can home damaged tissues. Although the homing mechanisms are not completely understood, their characteristics make them promising vehicles for target tumor therapy. On the other hand, the potential effects of MSC therapy on the growth and invasion of tumors is still controversial. MSCs have conflicting functions in modulating the tumor microenvironment, and can confer tumorigenic or anti-tumor potential to the tumor cells [83]. Indeed, MSCs can regulate tumor cell proliferation, angiogenesis, and metastasis through the secretion of EVs that control various cellular pathways and transfer signaling molecules [83]. The behavior of MSCs in clinical trials is still under investigation, but the idea of engineering their EVs has already emerged as another application in MSC-based cancer therapy [83].

In dogs, the effects of cAT-MSC CMs have been tested on in vitro models of melanoma [51] and hepatocellular carcinoma (HCC) [54]. Melanoma is the most common oral malignant tumor in dogs, characterized by high local invasiveness and metastatic propensity [84]. Based on the suggestion that apoptosis can be induced by the INF-β gene transfer in IFN-resistant tumors, such as melanoma, glioma, and renal cell carcinoma [85], cAT-MSCs were engineered through the lentiviral vector carrying the gene encoding canine INF-β (cINF-β) [51]. Regardless of the cINF-β expression, the CM from the cAT-MSCs inhibited LMeC melanoma cell growth through cell-cycle arrest and the induction of apoptosis, with a higher growth inhibitory effect for the increased cINF-β CM [51]. Thus, the cAT-MSC CM possesses an anti-proliferative effect on melanoma cells, which is enhanced by increased cINF-β production [51].

On the other hand, when the cAT-MSC CM was used on AZACH, a canine HCC cell line, it promoted cell proliferation and invasion in vitro [54]. Different concentrations (10%, 30%, and 50%) of the CM were used and the 30% concentration induced higher effects on the proliferation from day 2 of the culture [54]. The soluble factors contained in the CM showed different effects compared to those observed on human HCC cell lines, where the AT-MSC CM inhibited cell proliferation [86] or enhanced it but suppressed cell invasion [87]. After 48 h of culture, the canine HCC cells expressed higher levels of the TGF-β1, EGF (epidermal growth factor), HGF, PDGF-β (platelet-derived growth factor beta), VEGF-A, IGF-2 (insulin-like growth factor 2), MMP-2, and lower levels of MMP-9 [54]. The mRNA expression confirmed what was observed in the in vitro assays. In fact, TGF-β1 was associated with tumor cell progression, and the other growth factors also promoted HCC progression [54]. Moreover, MMP-2 and MMP-9 were involved in degrading type IV collagen and gelatine during tumor cell invasion and extracellular matrix degradation. In that study, the MMPs were differently regulated by the factors contained in the CM. Nonetheless, the high expressions of MMP-2 and MMP-9, in combination or alone, were highly correlated with HCC invasion and progression [54].

Additionally, for cAT-MSC secretome, controversial results were presented about its ability to inhibit or promote tumor progression and metastasis, depending on the type of tumor. The function of MSC secretome and the signaling pathways involved in various tumor microenvironments need to be elucidated to improve target therapy, and eventually customize it through cell engineering.

### 3.8. Other Applications

Infertility is a major concern in both human and veterinary medicine. Hormonal and surgical treatments, as well as assisted reproductive technologies, have been applied to reduce the impact of infertility. However, these disorders have overall health implications and long-term risks in humans [88], and high economic incidence in animals. Therefore, novel approaches and therapies need to be investigated. Regenerative medicine is a promising tool for the restoration and preservation of fertility. MSCs, through paracrine/autocrine mechanisms, possess regenerating, angiogenetic, and immune-modulating properties in addition to trophic, antioxidant, and anti-apoptotic actions. Recently, the use of MSCs, as well as their CMs and EVs, has been investigated for their potential in treating infertility [89,90] and improving sperm cryopreservation [91,92].

For sperm cryopreservation, the use of exosomes seems preferable to MSCs, as they do not increase the cell density. It has been hypothesized that the EVs derived from cAT-MSCs might stimulate the repair of sperm that is damaged during the freezing process [93]. Supplementation using EVs maintained the integrity of the plasma membrane as well as the chromatin material of canine sperm during cryopreservation [93]. Furthermore, it led to improvements in the standard indicators of sperm quality, including the progressive motility, amplitude of the lateral head displacement, and viability [93]. Considering the results obtained from the expression of the genes related to the repair of chromatin material and the plasma membrane (increased) and of those related to mitochondrial reactive oxygen species (decreased), it was evident that the addition of EVs improved the quality of post-thaw dog semen by initiating damaged sperm repair and decreasing reactive oxygen species production [93]. Further investigation into MSCs and their secretome is justified to explore their potential in sperm cryopreservation.

## 4. Conclusions

Although the studies on secretome derived from canine AT-MSCs are still limited, its characterization has confirmed similar features to those reported for secretions of MSCs from humans and other animal species. It contains EVs and soluble factors, including cytokines, growth factors, other immune mediators, and a number of proteins. In dogs, the use of cAT-MSC secretome, both as conditioned medium and EVs, in regenerative medicine appeared to be a valid alternative to cell–based applications, as some positive effects were observed in all the studies where secretome was applied. This potential is being explored in a range of applications, from addressing pathologies like diabetes and neurologic disorders, to tumor therapy, osteoarthritis, chronic infections, atopic dermatitis, wound healing, tympanic membrane repair, and sperm cryopreservation.

The use of secretome offers several advantages over more conventional cell–based approaches, including ease of handling, manufacturing, storage, and avoiding potential issues related to cell transplantation. A veterinary clinical grade cAT-MSC secretome production has been demonstrated, making it possible to develop a ready-to-use product, which can be stored for extended periods without compromising its quality and efficacy. Moreover, the composition of cAT-MSC secretome can be successfully modified by priming the cells to achieve the desired effects for both CMs and EVs. Nevertheless, further research is required to fully characterize and standardize the production of cAT-MSC secretome and its contained EVs.

## Figures and Tables

**Figure 1 animals-13-03571-f001:**
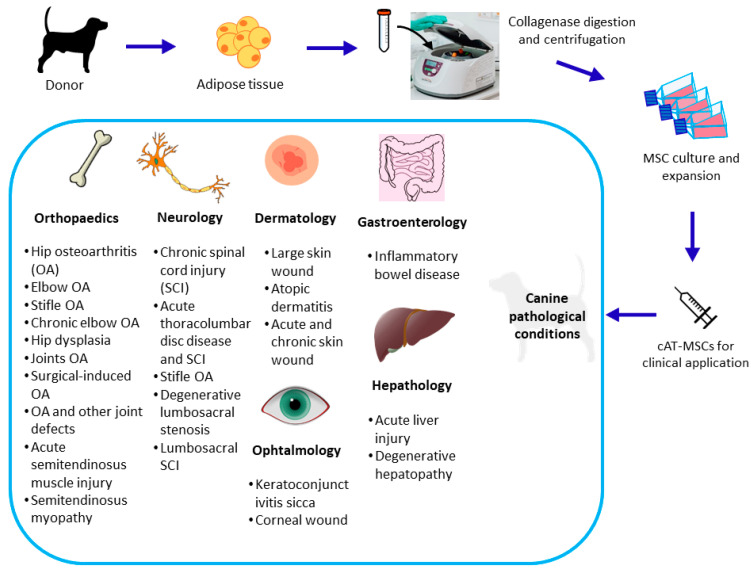
Graphical representation of the isolation and clinical applications of canine adipose tissue-derived mesenchymal stem/stromal cells (cAT-MSCs).

**Figure 2 animals-13-03571-f002:**
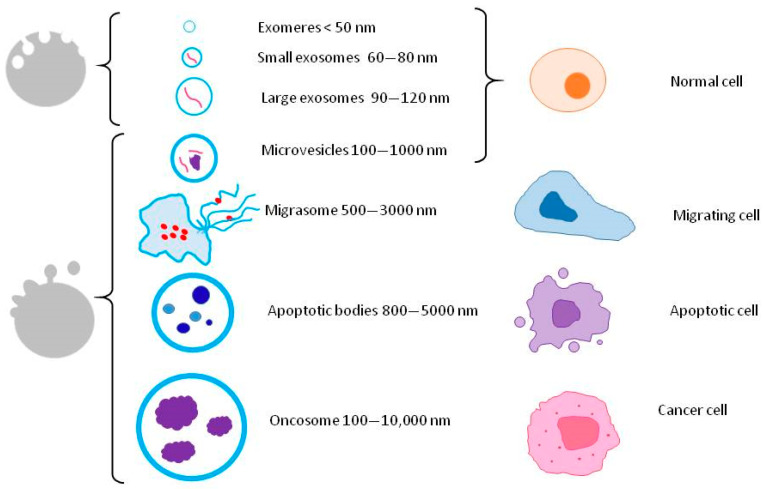
Classification of extracellular vesicles (EVs) according to their size and cell of origin. Different from the other subtypes of EVs, exomeres are defined as small (<50 nm), non-membranous extracellular nanoparticles. Exosomes originate as intraluminal vesicles while other EVs derive from the outward budding of the plasma membrane.

**Figure 3 animals-13-03571-f003:**
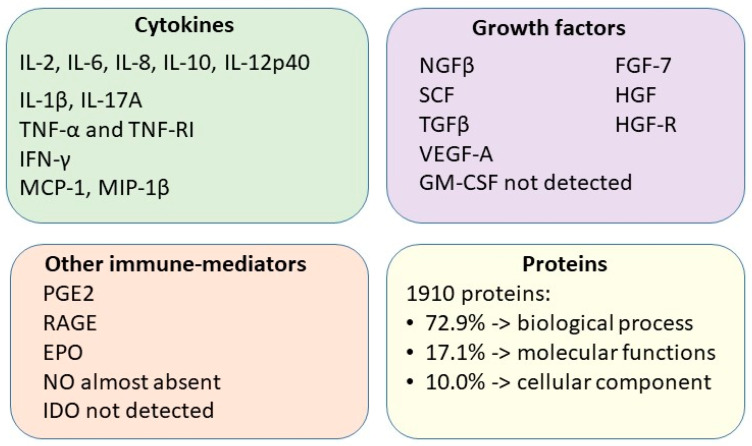
Representation of the identified soluble factors of secretome from canine adipose tissue-derived mesenchymal stem/stromal cells.

**Table 1 animals-13-03571-t001:** Isolation and characterization of extracellular vesicles derived from canine adipose tissue-derived mesenchymal stem/stromal cells.

Means	Isolation	Characterization	Reference
DMEM + 10% FBS exosomes free, 72 h	Centrifugation (13,000× *g* 30 min) and ultracentrifugation (100,000× *g* 60 min at 4 °C twice)	TEM, WB, proteomic analysis, and PBMCs proliferation in vitro test	[29]
DMEM + 10% FBS exosomes free, 72 h	Centrifugation (300× *g* 10 min at 4 °C + 2000× *g* 30 min at 4 °C), filtration (0.22 μm), ultracentrifugation (110,000× *g* 80 min at 4 °C twice), and filtration (0.22 μm)	Protein concentration, TEM, WB, and LPS-stimulated macrophage in vitro test	[30]
DMEM + 10% FBS exosomes free, 48 h	Centrifugation (300× *g* 10 min at 4 °C + 2000× *g* 30 min at 4 °C), filtration (0.22 μm), ultracentrifugation (110,000× *g* 80 min at 4 °C twice), and filtration (0.22 μm)	Protein concentration, WB, and TEM	[31]
DMEM + 10% FBS exosomes free, 48 h	Centrifugation (2600× *g* 20 min) + ExoQuick™(System Biosciences, Palo Alto, CA, USA) isolation according to the manufacturer’s instructions	WB, TEM, and LPS-stimulated macrophage in vitro test	[32]
DMEM + 10% FBS exosomes free, 72 h	MagCapture Exosome Isolation Kit PS: centrifugation (300× *g* 5 min + 1200× *g* 20 min at 4 °C), EVs blocking reagent + ultrafiltration (Vivaspin), and exosomes isolation according to the manufacturer’s instructions	NTA, TEM, protein concentration, miRNA PCR array, and PBMCs-based in vitro tests	[33]
DMEM/F12 serum free, 48 h	Centrifugation (3500× *g* 10 min), tangential flow filtration (5 kDa cut-off), diafiltration (with ultrapure water), and freeze-drying (+0.5% mannitol)	NTA	[34]
DMEM/F12 serum free, 48 h	Filtration (0.22 μm), tangential flow filtration (500 kDa cut-off), and diafiltration (with PBS)	NTA, WB, and bead-based FC	[35]
DMEM/F12 serum free, 48 h	Centrifugation (2000× *g* 10 min), filtration (0.22 μm), tangential flow filtration (500 kDa cut-off), and diafiltration (with DPBS)	NTA, protein concentration, bead-based FC, calnexin concentration, and miRNA profiling by NGS	[36]

DMEM = Dulbecco’s minimum essential medium; DPBS = Dulbecco’s phosphate-buffered saline; F12 = nutrient mixture F-12; FBS = fetal bovine serum; FC = flow cytometry; LPS = lipopolysaccharide; miRNA = micro RNA; NGS = next-generation sequencing; NTA = nanoparticle tracking analysis; PBMCs = peripheral blood mononuclear cells; PBS = phosphate-buffered saline; PCR = polymerase chain reaction; PGE2 = prostaglandin E2; TEM = transmission electron microscopy; WB = Western blot.

**Table 2 animals-13-03571-t002:** Applications of secretome derived from canine adipose tissue-derived mesenchymal stem/stromal cells.

Treatment	Disease (Type of Study)	Effect	Reference
CM	Melanoma (IV)	↑ apoptosis and ↓ growth of canine melanoma cells through IFN-β (overexpressed)	[51]
CM	Neurological disorders (IV)	↑ proliferation of SH-SY5Y neuronal cell line, neurite outgrowth, and immunopositivity for βIII-tubulin; ↑ proliferation and migration of EA.hy926 endothelial cell line; ↑ rate of wound closure in scratch assay	[52]
CM	Diabetes (IV)	↑ IRS-1 and GLUT4 in the insulin resistance model through FGF-1	[53]
CM	Hepatocellular carcinoma (IV)	↑ proliferation and invasion of canine HCC cells	[54]
EVs	Inflammatory bowel disease (IVO)	↓ DSS-induced colitis in murine model through TNF-α and IFN-γ (primed EVs)	[30]
EVs	Inflammatory bowel disease (IVO)	↓ DSS-induced colitis in murine model through TSG-6	[31]
CM	Spinal cord injury (IV)	↑ neuronal cell proliferation, neurite outgrowth, and βIII tubulin immunopositivity; ↑ endothelial cell migration, proliferation, and formation of tubule-like structures in Matrigel assay	[55]
CM	Osteoarthritis (IVO)	≠ MMP-3, TIMP-1, IL-6 and TNF-α levels in synovial fluid; ↑ ROM	[56]
CM	Osteoarthritis (IVO)	Lyosecretome safety confirmation (no relevant adverse response)	[34]
CM	Spinal cord injury (EV)	↑ glial cell survival and reactivity in SCSC; ↓ neuronal survival	[57]
CM	Chronic infections (IV)	↑ M2 phenotype of macrophage (resting MSC CM); ↑ macrophage bactericidal activity (activated MSC CM)	[58]
EVs	Atopic dermatitis (IVO)	Improvement of AD-like dermatitis in a mouse model: ↓ serum IgE, epidermal inflammatory cytokines and chemokines; ↑ skin barrier repair; ↓ pruritus	[35]
CM	Wound healing	No effect on second intention wound healing in turtles (*Trachemys scripta*)	[59]
CM	Ear diseases (IVO)	↑ tympanic membrane epithelial migration	[60]
EVs	Atopic dermatitis (IVO)	Improvement of AD-like dermatitis in a mouse model: ↓ serum IgE, inflammatory cytokines and chemokines; ↓ ear thickness	[36]

AD = atopic dermatitis; CD4 = cluster of differentiation 4; CM = conditioned medium; DFO = deferoxamine; DSS = dextrane sulfate sodium; EV = ex vivo; EVs = extracellular vesicles; FGF-1 = fibroblast growth factor 1; GFAP = glial fibrillary acidic protein; GLUT4 = insulin-regulated glucose transporter; HCC = hepatocellular carcinoma; IFN-β = interferon beta; IFN-γ = interferon gamma; IL-6 = interleukin 6; IRS-1 = insulin receptor substrate 1; IV = in vitro; IVO = in vivo; LPS = lipopolysaccharide; M1 = pro-inflammatory macrophage phenotype; M2 = anti-inflammatory macrophage phenotype; MMP-3 = matrix metalloproteinase-3; ROM = range of motion; SCSC = spinal cord slide culture; STAT3 = signal transducer and activator of transcription 3; TIMP-1 = tissue inhibitor metallopeptidase 1; TNF-α = tumor necrosis factor alpha; TSG-6 = TNF-α-stimulated gene/protein 6.

## Data Availability

No new data were created or analyzed in this study. Data sharing is not applicable to this article.

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
