# Peer review of "Beyond Canine Adipose Tissue-Derived Mesenchymal Stem/Stromal Cells Transplantation: An Update on Their Secretome Characterization and Applications"

_animals, 2023, doi:10.3390/ani13223571_

Round 1

Reviewer 1 Report

Comments and Suggestions for Authors

The research design and manuscript are promising. However, further research is needed to demonstrate its validity

Author Response

The manuscript is a review article. It has been modified as suggested by other reviewers.

Reviewer 2 Report

Comments and Suggestions for Authors

The manuscript aimed to review on state-of-the-art characterization and applications of canine adipose tissue derived MSCs secretome. The dog is a valuable animal model and concomitantly a pet for which advanced therapies are increasingly in demand. Moreover, MSCs release bioactive molecules, generally named secretome, which exert therapeutic functions. The secretome consists of many different soluble (such as growth factors, chemokines, and cytokines) and non-soluble factors (extracellular vesicles). Some specific suggestions I present below regarding the manuscript:

1. Figure 1 could be better elaborated. It seems unclear to me. I suggest authors organize it better and bring more information in it.

2. In table 1, I suggest changing 'source" to "means".

3. In Figure 3, delete “Graphical”.

4. In 'Conclusions', I suggest re-writing we better address the role of secretomes in regenerative medicine. Furthermore, I suggest contextualizing the advances observed in the canine species.

Author Response

The maniscript has been modified as suggested

Reviewer 3 Report

Comments and Suggestions for Authors

Dear Authors,

The manuscript by Merlo and Iacono, gives a comprehensive overview of the current landscape the canine secretome, the components and clinical applications.

I think it would be of great value to the present manuscript to add some practical considerations addressing the following aspects:

* Can you please indicate the number of MSCs that would be needed in average for expansion to obtain a therapeutically relevant quantity of EVs?

* In terms of the characterisation of canine EVs, have mostly established human markers been evaluated, are there some that are canine-specific - is there a need to have a species-specific approach? Could you please elaborate your opinion on this.

* You have mentioned briefly in the conclusions about some crucial differences between cell and cell-free therapies, also from the perspective of manufacturing, storing etc. Could you please expand this part maybe add one part in the introduction where you would highlight the strengths of cell-free therapy.

Comments on the Quality of English Language

Dear Authors,

The quality of English language is high. I have encountered small mistakes such as like 87 "made up" change to "made of", please revise the manuscript for this and double check the prepositions.

In terms of expressions, it would be perhaps more appropriate to maintain a neutral tone in the writing, for example in line 48 to change ''amazing'' to ''potentially valuable'', please revise the manuscript from this perspective.

Author Response

The manuscript has been modified as suggested
